# A De Novo 8q22.2q22.3 Interstitial Microdeletion in a Girl with Developmental Delay and Congenital Defects

**DOI:** 10.3390/medicina59061156

**Published:** 2023-06-15

**Authors:** Ruta Kalinauskiene, Deimante Brazdziunaite, Neringa Burokiene, Vaidas Dirsė, Ausra Morkuniene, Algirdas Utkus, Egle Preiksaitiene

**Affiliations:** 1Faculty of Medicine, Vilnius University, 01513 Vilnius, Lithuania; r.kalinauskiene@nhs.net; 2Department of Human and Medical Genetics, Institute of Biomedical Sciences, Faculty of Medicine, Vilnius University, 01513 Vilnius, Lithuania; deimante.brazdziunaite@santa.lt (D.B.); algirdas.utkus@mf.vu.lt (A.U.); 3Clinic of Internal Diseases and Family Medicine, Institute of Clinical Medicine, Faculty of Medicine, Vilnius University, 01513 Vilnius, Lithuania; neringa.burokiene@santa.lt; 4Hematology, Oncology and Transfusion Medicine Center, Vilnius University Hospital Santaros Klinikos, 01513 Vilnius, Lithuania; vaidas.dirse@santa.lt; 5Centre for Medical Genetics, Vilnius University Hospital Santaros Klinikos, 01513 Vilnius, Lithuania; ausra.morkuniene@santa.lt

**Keywords:** 8q22.2q22.3 microdeletion, intellectual disability, radioulnar synostosis

## Abstract

*Background and Objectives:* Only nine patients with interstitial de novo 8q22.2q22.3 microdeletions have been reported to date. The objective of this report is to present clinical features of a new patient with an 8q22.2q22.3 microdeletion, to compare her phenotype to other previously reported patients, and to further expand the phenotype associated with this microdeletion. *Materials and Methods:* We describe an 8½-year-old girl with developmental delay, congenital hip dysplasia, a bilateral foot deformity, bilateral congenital radioulnar synostosis, a congenital heart defect, and minor facial anomalies. Results: Chromosomal microarray analysis revealed a 4.9 Mb deletion in the 8q22.2q22.3 region. De novo origin was confirmed by real-time PCR analysis. *Conclusions:* Microdeletions in the 8q22.2q22.3 region are characterized by moderate to severe intellectual disability, seizures, distinct facial features and skeletal abnormalities. In addition to one already reported individual with an 8q22.2q22.3 microdeletion and unilateral radioulnar synostosis, this report of a child with bilateral radioulnar synostosis provides additional evidence, that radioulnar synostosis is not an incidental finding in individuals with an 8q22.2q22.3 microdeletion. Additional patients with similar microdeletions would be of a great importance for more accurate phenotypic description and further analysis of the genotypic-phenotypic relationship.

## 1. Introduction

To date, nine patients with an interstitial 8q22.2q22.3 microdeletion have been described in the literature [1,2,3,4,5]. The age of these patients ranged from early childhood to middle adulthood. Both males and females were affected. The size of the 8q22.2q22.3 microdeletions reported in the literature to date ranged from 1.36 Mb to 6.44 Mb. All nine patients were confirmed to have de novo microdeletions. The interstitial deletion phenotype has been described to involve moderate to severe intellectual disability, seizures, microcephaly, short stature, and a distinct facial phenotype, including minor anomalies of the eyes and down-turned corners of the mouth.

Here we describe a girl with developmental delay, congenital anomalies, and a de novo 4.9 Mb 8q22.2q22.3 deletion, and provide additional data for further delineation of this rare chromosomal alteration.

## 2. Materials and Methods

### 2.1. Clinical Evaluation

The girl, 8½ years of age (Decipher 339422), was a second child, born to healthy non-consanguineous Lithuanian parents. At 32 weeks of gestation, a fetal ultrasound showed mild growth retardation. The proband was born at 38 weeks of gestation via spontaneous vaginal delivery. Her birth weight was 2670 g (below the 3rd centile), her length was 50 cm (25th centile), her occipitofrontal circumference was 31 cm (below 3rd centile), and her Apgar score was 10. Hearing screening was negative for the left ear in infancy, but further investigation and a BERA test showed no hearing pathology. The phenotype was remarkable for up-slanting palpebral fissures, epicanthus, wide eyebrows, narrow ear canal, altered dermatoglyphics, short halluces, and increased hair growth on the back, arms and forehead (Figure 1). Skeletal abnormalities included congenital hip dysplasia, bilateral rotational deformity of the feet, and bilateral congenital radioulnar synostosis, confirmed by radiological investigation (Figure 1). An osteotomy of the synostosis of the right forearm, fixing it in a neutral position, was performed at age of 6 years. Her brain and renal ultrasound scans were normal. An echocardiogram showed open foramen ovale and a small secondary atrial septal defect. The echocardiogram repeated at age 3½ years was normal. She was also noticed to have stereotypic movements, including hand waving and head twisting, at age of 2½ years. She also had a single seizure episode with no epileptic activity on EEG. She developed headaches at age 7 years. Standard sleep EEG showed two episodes of generalized epileptiform discharges and one episode of dysrhythmia, but no specific antiepileptic treatment was started at the time. Brain MRI showed a small arachnoid cyst in the left temporal region. Other problems involved strabismus, hypermetropia, atopic dermatitis and food allergy for cow’s milk, wheat, rice, rye, and barley. Our patient presented with global developmental delay. She crawled at 18 months and walked without support at 2 years of age. A DISC (Dominance, Influence, Steadiness, Compliance) personality test at age 32 months was in the range of 47% for expressive language to 50% for gross motor skills. Her psychological evaluation at age of 6 years yielded a verbal scale of 60, a performance of 50 and a full scale of 51 on the Wechsler Intelligence Scale for Children (WISC-III). During clinical assessment at the age of 2.5 years, her head circumference was 46 cm (3rd centile), her height was 88 cm (25th centile) and her weight was 12 kg (25th centile). At the age of 8.5 years her head circumference was 51.5 cm (10–25th centile), her height was 133 cm (50th centile) and her weight was 35 kg (90th centile), the facial minor anomalies remained as described after birth. 

### 2.2. Chromosomal Microarray Analysis

The patient’s DNA sample was investigated using the Infinium HD whole-genome genotyping assay with the HumanCytoSNP-12 BeadChip (Illumina Inc., San Diego, CA, USA), which covers the entire genome with an average spacing of 9.6 kb and allows an average resolution of 31 kb. Sample was processed and the assay was performed according to a routine protocol provided by the manufacturer. Genotypes were called by GenomeStudio GT module version 1.7 (Illumina Inc.), and further analyzed with QuantiSNP version 1.1 and KaryoStudio version 1.0.3 software (Illumina Inc.). Constitutional copy number polymorphisms were excluded based on comparison with the Database of Genomic Variants (http://projects.tcag.ca/variation; accessed on 1 November 2022).

### 2.3. Real-Time PCR Analysis

Real-time PCR analysis was performed for the proband and both parents to confirm and determine the origin of the deletion previously detected by chromosomal microarray analysis. Two different pairs of specific primers located in the deleted region were used in real-time PCR. The first pair was located in ZNF706 (MIM#619526) gene (forward primer: 5’ GGCTCGTGGACAGCAGAAA-3’; reverse primer: 5’ CCCCTTACCCTACAGACAGTGC-3’), and the second pair was located in SNX31 (MIM#619839) gene (forward primer: 5’ GGCACTATGGATACCTGCAGCT-3’; reverse primer: 5’ GCACTTCACCCTGCTCATCTG-3’). Additionally, one pair outside the deletion in ABRA (MIM#609747) gene (forward primer: 5’ CCTCAGCCACAGGTACGAGA-3’; reverse primer: 5’ TCCATCACTCTCCAGCCCTTG-3’) was used. Primers were created using the Primer3 program (http://frodo.wi.mit.edu/cgibin/primer3/primer3_www.cgi; accessed on 1 November 2022). Real-time PCR reactions were performed in triplicate using 20 ng of genomic DNA in a 25-μL total volume, containing 2.5 pmol of each primer and SYBR^®^ Green PCR Master Mix (Applied Biosystems, Foster City, CA, USA). Samples were subjected to the initial step at 50 °C for 2 min, 40 cycles of denaturation (95 °C, 15 s), and combined annealing and extension (60 °C for 1 min). The Real-time PCR reactions were performed using the ABI Prism 7900HT Sequence Detection System and associated SDS software v2.3 (Applied Biosystems). Comparative Ct Method (ΔΔCt) for relative quantitation was applied, together with normalization to an endogenous control. As an endogenous control or as normalizer gene MED18 (MIM#612384) gene was used (forward primer: 5’ TGCGAAACTGCGTGGACATT-3’; reverse primer: 5’ ATGCGGAAGCCCATTTCCAT-3’).

## 3. Results

Chromosomal microarray analysis revealed a 4.9 Mb deletion in the 8q22.2q22.3 region, arr[GRCh38] 8q22.2q22.3(98926242_103924318) × 1 (Figure 2). Real-time PCR analysis of the proband and her parents confirmed the deletion in the proband and revealed its de novo origin.

## 4. Discussion

Interstitial deletions in the 8q22.2q22.3 region were first described in 2011 by Kuechler et al. [1]. The authors reported five patients with overlapping microdeletions and similar clinical features, including moderate to severe developmental delay, seizures, microcephaly, short stature, minor facial anomalies, and skeletal abnormalities. Kuroda et al. described an additional patient (Patient 6) with intellectual disability, epilepsy and short stature, to date the smallest (1.36 Mb) deletion in the 8q22.3 region [2]. The oldest patient, a 40-year-old male (Patient 7) with a 3.35 Mb microdeletion, in addition to intellectual disability, seizures, and specific facial features, had hearing loss and congenital heart disease, which had not been described in individuals with 8q22.2q22.3 deletions before. The authors proposed the GRHL2 gene as a possible candidate for the patient’s hearing loss, given its known association with autosomal dominant non-syndromic hearing loss [3]. The reported 10-year-old girl (Patient 8) presented with a phenotype indicative of the 8q22.2q22.3 microdeletion, including intellectual disability, epilepsy, short stature, microcephaly, specific facial features, hearing loss, and congenital heart disease [4]. The youngest patient, a 2-year-old girl (Patient 9) exhibiting most of the characteristic features of 8q22.2q22.3 microdeletion, was recently reported by Sharaf-Eldin et al. [5]. The deletion in 8q22.2, 1.027 Mb in size, was detected in a patient with isolated occult subtotal cleft of the secondary palate and no specific anomalies for the 8q22.2q22.3 microdeletion syndrome; this deletion was therefore not included for further analysis [6]. The 4.9 Mb deletion detected in our patient overlaps with deletions seen in all patients described in the literature to date (Figure 2). The deleted region in our patient encompasses 75 genes; 28 of them are protein coding genes and 9 are disease-associated genes (*VPS13B*, *FBXO43*, *SPAG1*, *GRHL2*, *RRM2B*, *FZD6*, *CTHRC1*, *SLC25A32*, and *RIMS2*, of which only *GRHL2* and *RRM2B* are associated with autosomal dominant phenotypes, deafness and progressive external ophthalmoplegia with mitochondrial DNA deletions, respectively) (Figure 2).

A summary of the detailed clinical features of previously reported individuals with interstitial deletions in 8q22.2q22.3 and our patient is shown in Table 1. All ten individuals were born to healthy non-consanguineous parents at gestation ages between 37 and 42 weeks. All patients presented with developmental delay and/or moderate to severe intellectual disability. Most of these patients had speech impairment and various forms of autistic behavior. Seizures manifested in nine out of ten patients with a variation from the absences to generalized tonic-clonic and grand mal seizures. The girl with a smallest reported deletion in the 8q22.3 region described by Kuroda et al. had both intellectual disability and early onset epilepsy. The minimal overlapping region of all patients with seizures was 460 kb in size (genomic coordinates 8:102020269-102480673), encompassing three genes (*RRM2B, UBR5* and the proximal part of *NCALD*). *NCALD* and *UBR5* are likely to be dosage sensitive, with pLI scores 0.89 and 1, respectively, but there are no known associated phenotypes with these genes. *UBR5* encodes a progestin-induced protein and potentially has a role in the regulation of cell proliferation or differentiation [7]. *NCALD* encodes a neuronal calcium sensor protein that is involved in calcium signaling [8]. *NCALD* is highly expressed in cerebral neurons, spinal motor neurons, axonal growth cones, and at the presynaptic terminals of the neuromuscular junction and plays an important role in the regulation of the neuronal signal transduction process [9]. However, the chromosomal alteration detected in seizure-free Patient 2 overlaps this chromosomal region and this may suggest more complex etiology, incomplete penetrance, or later manifestation of seizures, as Patient 2 was only 3.5 years old at the time of publication.

Congenital or postnatal microcephaly was observed in eight of the ten patients. Patient 7 was the only one with notable macrocephaly, though this might be a familial trait, not related to his microdeletion [3]. In addition to microcephaly, seven patients were diagnosed with antenatal growth restriction (IUGR), which resolved in our patient after birth, while progressing in the remaining five patients. Postnatal short stature developed in Patient 2 and Patient 8. Although there was a tendency for the 8q22.2q22.3 microdeletion to be associated with short stature, it might be worth re-evaluating the reported patients later in life, since almost all were of a relatively young age during clinical assessment.

Eight of the ten patients had minor anomalies of the periorbital region, including blepharophimosis, ptosis, telecanthus, epicanthus, up-slanting palpebral fissures, and sparse eyebrows. Other reported facial anomalies included flat nasal tip, thin upper lip vermilion, down-turned corners of the mouth, ear abnormalities, and a poor facial movement or little facial expression. The smallest overlapping region for characteristic periorbital anomalies was 1.3 Mb in size (8:99607311-100935752) and involved eleven protein coding genes, two of which (*PABPC1* and *YWHAZ*) have high pLI scores. The associations of these genes with diseases are unknown, however.

Our patient was diagnosed with a congenital bilateral radioulnar synostosis, which was also reported in Patient 3 [1]. It is therefore likely that this rare skeletal abnormality is not an incidental finding in our patient. This further expands the spectrum of clinical manifestations of the 8q22.2q22.3 microdeletion. Congenital radioulnar synostosis is a rare abnormality caused by the failure of the fused cartilaginous precursors of the radius and ulna to separate during the seventh week of gestation [10]. Sixty to eighty percent of congenital radioulnar synostoses are bilateral and they affect male and female patients equally [11]. As reported by Tsai et al., this abnormality may be associated with other congenital musculoskeletal manifestations such as congenital hip dislocation, clubfeet, polydactyly, or syndactyly [12]. However, it can also be an isolated anomaly [13]. In addition to radioulnar synostosis, Patient 3 also had short hands with proximally implanted thumbs and mild cutaneous finger syndactyly, whilst our patient had congenital hip dysplasia, bilateral foot deformity, and short halluces. Skeletal abnormalities such as small hands and short thumbs and toes were also present in Patient 2 and Patient 8. The overlapping region of our patient and Patients 2, 3 and 8 was 3.86 Mb in size (8:99607311-103474851) and encompassed 25 protein coding genes. Since the region is too large for the identification of a single candidate gene responsible for skeletal involvement, further studies on the deleted region should therefore be considered.

Hearing loss, which was seen in two patients, may be associated with *GRHL2*, as this gene is a known cause of autosomal dominant non-syndromic sensorineural hearing loss. Vona et al. described homozygous *GRHL2* mutant zebrafish embryos with enlarged otocysts, thinner otic epithelium and smaller or eliminated otoliths, which proves the importance of this gene for hearing [14]. Loss-of-function variants in *GRHL2* typically result in progressive late onset hearing loss. For that reason, regular Audiology assessments should be recommended for individuals with deletions encompassing the *GRHL2* gene.

## 5. Conclusions

In conclusion, microdeletions in the 8q22.2q22.3 region are rare and have been reported in only nine unrelated individuals worldwide to date. An 8q22.2q22.3 microdeletion is characterized by moderate to severe intellectual disability, seizures, distinct facial features and skeletal abnormalities. In addition to one already reported individual with an 8q22.2q22.3 microdeletion and unilateral radioulnar synostosis, this clinical report of a child with bilateral radioulnar synostosis provides additional evidence, that radioulnar synostosis is not an incidental finding in individuals with an 8q22.2q22.3 microdeletion. Additional patients with similar microdeletions would be of a great importance for more accurate phenotypic delineation and further analysis of the relationship of genotype to phenotype. 

## Figures and Tables

**Figure 1 medicina-59-01156-f001:**
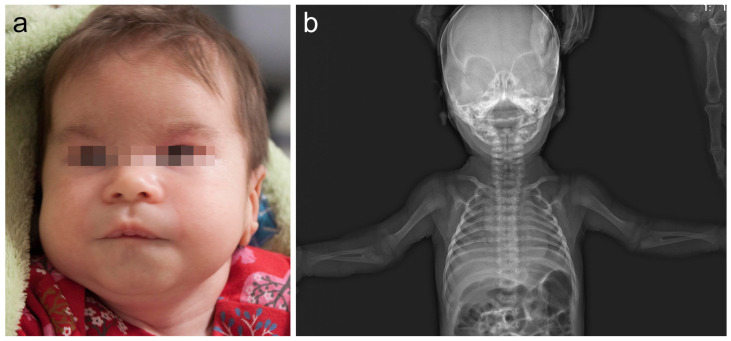
(**a**) The patient at 1.5 months of age. Note minor facial features: frontal bossing, wide eyebrows, depressed nasal bridge, short and wide nose, and down-slanting corners of the mouth. (**b**) Skeletal X-ray of our patient at 1 month of age revealed bilateral radioulnar synostosis.

**Figure 2 medicina-59-01156-f002:**
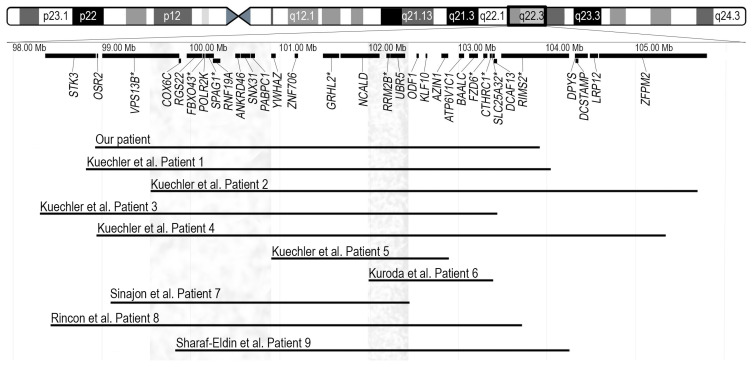
Ideogram of chromosome 8 and the schematic view of genes located in the highlighted chromosomal region are presented in the top portion. Disease associated genes are marked with an asterisk. Horizontal lines in the bottom represent the 4.9-Mb deletion arr[GRCh38] 8q22.2q22.3(98926242_103924318) × 1 detected in our patient and nine previously reported overlapping deletions. Coordinates of deletions are remapped to GRCh38 (hg38) assembly. Grey vertical bars in left and right show the chromosomal regions possibly responsible for periorbital features and seizures, respectively [1,2,3,4,5].

**Table 1 medicina-59-01156-t001:** Clinical features of reported affected individuals. Abbreviations: F—female; M—male; na—not available; DD/ID—developmental delay/intellectual disability; IUGR—intrauterine growth restriction; +—identified feature; −—feature not identified. Coordinates of deletions are remapped to GRCh38 (hg38) assembly.

Author	Kuechler et al. [1]					Kuroda et al. [2]	Sinajon et al. [3]	Rincon et al. [4]	Sharaf-Eldin et al. [5]	Our Patient
Patient No.	1	2	3	4	5	6	7	8	9	10
Deletion Range	98,813,948–104,069,980	99,607,311–105,707,857	98,212,392–103,474,851	98,943,820–105,387,943	100,935,752–102,858,169	102,020,269–103,427,344	99,130,152–102,480,673	98,428,025–103,801,718	99,958,13–104,284,284	98,926,242–103,924,318
Age at clinical assessment (years)	6	3½	8½	8	20	8	40	10	2	8½
Gender	F	M	F	F	F	F	M	F	F	F
DD/ID	+	+	+	+	+	+	+	+	+	+
Seizures	+	−	+	+	+	+	+	+	+	+
Congenital microcephaly	+	−	+	+	+	−	na	na	−	+
Microcephaly at time of evaluation	+	+	+	+	+	−	−	+	+	−
IUGR	+	-	+	+	+	+	−	-	+	+
Short stature	+	+	+	+	+	+	−	+	+	−
Minor anomalies of eyes	+	+	+	+	−	−	+	+	+	+
Down-turned corners of mouth	+	+	+	+	+	−	+	+	−	+
Short or proximally implanted thumbs/halluces	−	+	+	−	−	na	na	+	−	+
Radio-ulnar synostosis	−	−	+	−	−	−	−	−	−	+
Deafness	−	−	−	−	−	−	+	+	−	−
Congenital heart disease	−	−	−	−	−	−	+	+	−	+
Hiatal or diaphragmatic hernia	−	+	−	+	−	−	−	−	−	−

## Data Availability

The main data generated and analyzed during this study are included in this article. Any additional information is available from the authors upon request.

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
