# Peer review of "A De Novo 8q22.2q22.3 Interstitial Microdeletion in a Girl with Developmental Delay and Congenital Defects"

_medicina, 2023, doi:10.3390/medicina59061156_

Round 1

Reviewer 1 Report

I consider this work important in expanding what we know about the 8q22.2q22.3 microdeletion, especially about the phenotypic spectrum, and might be useful in the future for assessing the exact genotype-phenotype correlations.

Minor comments:

1.      It would be interesting to present facial dysmorphism at different ages.

2.      What type of foot deformities does the patient have?

Author Response

Point 1: It would be interesting to present facial dysmorphism at different ages.

Response 1: There were no remarcable changes in facial dysmorphism with age. We added this information in the section of clinical evaluation: „The facial minor anomalies remained as described after birth“.

Point 2: What type of foot deformities does the patient have?

Response 2: The patient has internal rotational deformity of the feet. This information was added in the section of clinical evaluation.

Reviewer 2 Report

The study is well written. However, an Egyptian patient with 8q22.2q22.3 microdeletion has been recently published in human gene journal (Sharaf-Eldin et al., 2022). The present study should be rewritten including the Egyptian case.

Is your patient Lithuanian? Please, clarify.

Author Response

Point 1: The study is well written. However, an Egyptian patient with 8q22.2q22.3 microdeletion has been recently published in human gene journal (Sharaf-Eldin et al., 2022). The present study should be rewritten including the Egyptian case.

Response 1: Thank you, we added the information about the article in Human gene journal (Sharaf-Eldin et al., 2022). We missed it, because this article was not in PubMed. Accordingly, we have updated the text, references, a table and figure 2.

Point 2: Is your patient Lithuanian? Please, clarify.

Response 2: Yes, both parents are Lithuanians. We added this information in Clinical evaluation section.

Reviewer 3 Report

  • Until recent only  8 patients with interstitial de novo 8q22.2q22.3 micro- 15 deletions have been reported. Article reports on is to present clinical features of new patient.
  • This case report presents patient with developmental delay, congenital hip dysplasia, bilateral foot deformity, bilateral congenital radioulnar synostosis, a congenital heart defect, and minor facial anomalies.
  • Authors describe congenital anomalies, they focus in whole article on bilateral radioulnar synostosis. Article is fluent and clear. 
  • Based on author´s detailed discussion on radioulnar synostosis update in literature, more recent publications on proximal radioulnar synostosis (PRUS) could be provided. It is on author´s consideration to cite recent work on PRUS from Medicina.

Author Response

Point 1:  Based on author´s detailed discussion on radioulnar synostosis update in literature, more recent publications on proximal radioulnar synostosis (PRUS) could be provided. It is on author´s consideration to cite recent work on PRUS from Medicina.

Response 1: As suggested, we cited more recent publication on PRUS.

Round 2

Reviewer 2 Report

Please, add add cow milk allergy to the table as it is currently reported in 2 patients. It would be important to report if an association could be existed between this allergy and the microdeletion itself or not.